# Anatomical Validation of a Selective Anesthetic Block Test to Differentiate Morton’s Neuroma from Mechanical Metatarsalgia

**DOI:** 10.3390/reports8040211

**Published:** 2025-10-21

**Authors:** Gabriel Camuñas-Nieves, Hector Pérez-Sánchez, Alejandro Fernández-Gibello, Simone Moroni, Felice Galluccio, Mario Fajardo-Pérez, Laura Pérez-Palma, Alfonso Martínez-Nova

**Affiliations:** 1Clínica Vitruvio, C/María de Guzmán 47, 28003 Madrid, Spain; gabrielcamunas@gmail.com (G.C.-N.); hperezsanchez00@gmail.com (H.P.-S.); alejandrofernandezgibello@gmail.com (A.F.-G.); 2Podiatry Department, San Vicente Mártir University, Pl. Almoina 3, 46003 Valencia, Spain; dott.simonemoroni@gmail.com; 3Fisiotech Lab Studio, Rheumatology and Pain Management, 50136 Firenze, Italy; felicegalluccio@gmail.com; 4Center for Regional Anesthesia and Pain Medicine (CRAPM), Wan Fang Hospital, Taipei Medical University, Taipei 11696, Taiwan; 5Ultradissection Group, 28029 Madrid, Spain; mfajardoperez@yahoo.es; 6Department of Clinic Sciences, University of Barcelona, 08907 l’Hospitalet de Llobregat, Spain; lperez@ub.edu; 7Department of Nursing, University of Extremadura, 10600 Plasencia, Spain

**Keywords:** Morton’s neuroma, metatarsalgia, ultrasound-guided injection, differential diagnosis, peripheral neuropathy

## Abstract

**Background and Objectives:** The anesthetic nerve block test is a surgical technique that can assist in the differential diagnosis of forefoot pain. The MTP joint, enclosed by its capsule, may act as a sealed cavity with predictable contrast dispersion, whereas the IM space, lacking clear boundaries and containing bursae and the plantar digital nerve, favors diffuse spread. Due to the high rate of false positives in suspected cases of Morton’s neuroma with the anesthetic block current procedure in the intermetatarsal space, the aim of this study was to propose an alternative to the current procedure. **Material and Methods**: Six fresh cadaveric feet were used. Under ultrasound guidance, the 2nd–4th MTP joints received stepwise intra-articular injections of radiopaque contrast. The third common digital nerve was injected within the third intermetatarsal space. Standard radiographs were obtained to assess distribution and proximal spread. **Results:** A volume of 0.3 mL was sufficient to fully reach the intra-articular cavity and potentially induce effective localized anesthesia. When the third common digital plantar nerve was injected in an anatomically healthy region, the contrast medium showed a proximal diffusion pattern extending up to the mid-diaphyseal level of the third and fourth metatarsal bones. On radiographs, the intra-articular infiltration lines appear sharply demarcated, supporting the interpretation of the metatarsophalangeal joint as a sealed compartment. **Conclusions:** Low intra-articular anesthetic volumes may yield targeted effects, while Morton’s neuroma injections spread proximally, risking loss of diagnostic specificity; this technique may improve decision-making accuracy and reduce failures.

## 1. Introduction

Metatarsalgia is one of the most prevalent causes of forefoot pain, with an estimated incidence of approximately 10% in the general population, affecting females more frequently [1]. It is a multifactorial clinical syndrome characterized by plantar pain in the region of the metatarsal heads, whose etiology may involve structural, mechanical, neurological, or inflammatory alterations. One of the conditions associated with metatarsalgia is Morton’s neuroma, defined as a compressive neuropathy of the common plantar digital nerve, with a higher prevalence in the third intermetatarsal space. Clinically, it presents as a burning pain radiating to the adjacent toes, often accompanied by paresthesias, electric shock-like sensations, and exacerbation when wearing tight footwear. Its pathogenesis involves perineural fibrosis, axonal degeneration, and degenerative vascular changes, potentially resulting from chronic repetitive trauma to the forefoot and biomechanical factors such as compression by the deep transverse intermetatarsal ligament [2,3,4].

Nevertheless, the diagnosis of Morton’s neuroma should not be established without considering other causes of forefoot pain, including mechanical overload of the metatarsal heads [2,4], plantar plate pathology [5,6], bursitis or synovitis [7] stress fractures [8,9], neuropathies [4], or inflammatory arthritides such as rheumatoid arthritis [10,11], which may present with overlapping symptoms and require different therapeutic approaches. This diagnostic complexity underscores the importance of anatomical-clinical correlation in distinguishing articular from neuropathic pain [2,4].

The differential diagnosis of forefoot pain is challenging because several anatomical structures—including MTP joints, plantar digital nerves, and periarticular tissues—can produce overlapping symptoms. Correctly identifying the pain generator is essential, as treatment would differ substantially: plantar plate rupture might require reconstructive surgery or offloading orthoses, while Morton’s neuroma would be more amenable to anesthetic injections, radiofrequency ablation, or neurectomy [2,12]. This clinical overlap highlights the potential importance of functional diagnostic tools grounded in anatomy, which would complement imaging findings by distinguishing intra-articular from neuropathic pain [12,13].

From an anatomical standpoint, the studies by Sarrafian and Kelikian provide a foundational description of the neural and articular arrangement of the forefoot [12]. The MTP joints are innervated by dorsal and plantar articular branches arising from the superficial and deep peroneal nerves, as well as the medial and lateral plantar nerves. These articular branches emerge proximal to the formation of the common plantar digital nerve, implying that an intra-articular anesthetic injection should not block a neuroma located distally in the intermetatarsal space. This functional separation is key to understanding the diagnostic utility of selective injections [12]. Regarding available diagnostic tools, ultrasound and magnetic resonance imaging (MRI) are the most frequently employed imaging modalities [9], with fluoroscopy also demonstrating good results [14]. Ultrasound demonstrates high sensitivity for detecting interdigital nerve thickening and associated bursitis, with the added advantage of enabling dynamic and bilateral comparative assessment. However, its specificity is limited by the presence of nonspecific findings in asymptomatic individuals and its operator dependency. MRI, on the other hand, offers superior visualization of the integrity of the plantar plate, bursae, synovitis, or stress fractures, although it has a high false-positive rate for neuromas, particularly in studies lacking clinical correlation [15]. To overcome these limitations, several functional tests have been developed, such as ultrasound-guided diagnostic anesthetic blocks. These procedures involve the selective administration of local anesthetic into specific regions of the forefoot—such as the MTP joint or the neuroma site—followed by clinical evaluation of symptom relief. If intra-articular injection alleviates pain, an articular source is confirmed; if not, a neuropathic origin is more likely. The literature supports these tests as highly predictive diagnostic tools, validated in the knee, hip, lumbar spine, and more recently in the foot, as demonstrated by Ruiz Santiago et al. [3] and El-Khoury et al. [16].

Currently, the nerve suppression test is performed by injecting local anesthetic into the intermetatarsal space, directing the needle from dorsal to plantar, perpendicular to the course of the common digital plantar nerve. This technique aims to block Morton’s neuroma, so that if the patient experiences relief, the pain is considered to be of neuropathic origin [3,16,17]. However, this approach may present a high rate of false positives, as observed in our experimental series. The contrast injection demonstrated a wide proximal spread of the anesthetic along the course of the lateral plantar nerve, reaching the plantar articular branches of the metatarsophalangeal joint capsule. This phenomenon implies that the clinical improvement after infiltration may not be due to the effect on the neuroma itself, but rather to the unintentional blockade of neighboring intra-articular structures.

Alternatively, we propose an opposing technique, based on ultrasound-guided diagnostic intra-articular infiltration of the metatarsophalangeal joint, a sealed cavity consistent with its capsular anatomy, that would allow the anesthetic effect to remain confined to the articular compartment. The clinical rationale for this approach relies on the proximal origin of the MTP articular branches, as described in classical anatomical studies [13]. If the patient does not experience improvement after this infiltration, it may suggest that the pain does not originate from the joint and that the most likely cause is neuropathic. This sequential strategy—first infiltrating the joint and, only in the absence of response, performing the nerve suppression test—could improve diagnostic specificity and reduce the risk of misinterpretation. In addition, the use of ultrasound ensures anatomical precision and technical safety [3].

This type of strategy may have the potential to extend beyond diagnostic application, as it could also serve as a predictor of therapeutic response. For instance, patients experiencing pain relief after neuroma infiltration might be considered candidates for percutaneous radiofrequency ablation or selective neurotomy, whereas a positive response to joint blockade could indicate the need for focal orthopedic procedures. In this context, anatomical-clinical and radiological correlation—as explored in this study through contrast-enhanced dispersion analysis—may provide a more objective interpretation of pain origin and support more individualized treatment planning. The aim of this study was therefore to propose an alternative diagnostic approach and to evaluate its feasibility through radiographic results in cadaveric specimens.

## 2. Materials and Methods

An experimental and descriptive anatomical study was conducted, inspired by validated protocols described by Ruiz Santiago et al. [3], with the objective of documenting the radiological dispersion of iodinated contrast medium following ultrasound-guided injection into two key regions of the forefoot: the metatarsophalangeal (MTP) joint and the intermetatarsal (IM) space. This approach allows for anatomical validation of the clinical rationale behind the suppression test used to differentiate between Morton’s neuroma and other sources of metatarsalgia.

### 2.1. Anatomical Specimens and Inclusion Criteria

Six fresh human lower limbs were used, donated to institutional educational and research programs. All specimens were previously examined to confirm the anatomical integrity of the forefoot and to rule out signs of prior surgery, trauma, or deformities that could interfere with the technique.

### 2.2. Ultrasound-Guided Injection Technique

The procedure was performed using high-resolution ultrasound (10–15 MHz linear transducer) under real-time guidance (Figure 1 and Figure 2), following the selective infiltration protocol described by Ruiz Santiago et al. for Morton’s neuroma [3]. Three injections were performed on each anatomical specimen: 1—metatarsophalangeal joint of the third ray (3rd MTP joint), 2—metatarsophalangeal joint of the fourth ray (4th MTP joint), and 3—intermetatarsal space between the third and fourth rays (IM space).

A 25G, 40 mm needle was used. A volume of 1 cc of iodinated contrast medium (iopamidol 300 mg/mL) was injected at each site. For the MTP joints, a dorsal longitudinal approach was employed, with the needle oriented parallel to the metatarsal shaft and advanced to the base of the proximal phalanx, avoiding the dorsal capsule and extensor tendons. For the IM space, a transverse dorsoplantar approach was used. The needle was positioned in the deep plane of the intermetatarsal space, just beneath the deep transverse intermetatarsal ligament and above the course of the common plantar digital nerve.

The selected volume of 1 cc was based on previous studies [16,17] and personal clinical experience, which indicate that this amount provides sufficient visualization without risk of extensive extravasation or pressure-related artifacts that could distort radiological interpretation.

### 2.3. Radiological Assessment

Digital radiographs were obtained in two projections (dorsoplantar and anteromedial oblique) after each injection. Images were acquired using a standardized exposure system (55 kV, 2.5 mAs). Two independent observers (second and fourth authors) with expertise in musculoskeletal imaging analyzed the images in a blinded fashion, evaluating the following parameters: a—contrast dispersion: localized/intra-articular vs. diffuse/extra-articular, b—margin definition: well-defined vs. irregular, and c—relationship with anatomical compartments as described by Sarrafian [12]. Images were stored in DICOM format for subsequent review and were accompanied by static and video ultrasound recordings of each injection.

### 2.4. Quality Control and Reproducibility

To ensure replicability, the following quality control measures were implemented: All injections were performed by the same experienced operator. Each procedure was documented both sonographically and audiovisually. Two injections were randomly repeated to verify consistency of the dispersion pattern.

### 2.5. Ethical Considerations

All specimen donations complied with applicable legislation. This study was approved by the Research Ethics Committee of the University of Extremadura (ID 97//2022) and were carried out in accordance with institutional ethical protocols, aligned with the Declaration of Helsinki as adapted for cadaveric research.

## 3. Results

A total of 18 ultrasound-guided injections were performed across 6 fresh anatomical specimens: 6 into the third metatarsophalangeal (3rd MTP) joint, 6 into the fourth metatarsophalangeal (4th MTP) joint, and 6 into the third intermetatarsal (IM) space. All procedures were successfully executed under real-time ultrasound guidance and followed by standardized radiographic acquisition.

### 3.1. Injections into the Metatarsophalangeal Joints (3rd and 4th MTP)

In all intra-articular injections (n = 12), iodinated contrast dispersion remained confined within the articular capsule (Figure 3). Radiographic images revealed a contained, symmetrical pattern with sharply defined margins and an ovoid or triangular morphology, depending on the projection plane. The distribution of contrast was homogeneous in all joints, with no evidence of extravasation into the collateral ligaments, dorsal capsule, or pericapsular regions. These findings support the hypothesis that the MTP joint functions as a closed, sealed cavity, offering a significant anatomical advantage for performing selective and reproducible diagnostic blocks.

### 3.2. Injections into the Intermetatarsal Space (IM)

In all specimens, IM space infiltration resulted in diffuse, uncontained dispersion with irregular expansion and poorly defined borders (Figure 4). The contrast extended both proximally and laterally into adjacent soft tissue planes, following fascial paths with no clear anatomical containment. Additionally, a tendency for contrast migration toward the plantar region was observed, following the trajectory of the common plantar digital nerve. The mobility and superficial location of this nerve within the IM space appear to facilitate the lack of containment. This pattern confirms that the IM space functions as an anatomically open cavity with higher permeability to infiltrated fluid.

### 3.3. Functional Comparison of Dispersion Patterns

The results obtained in the experimental study revealed clear and consistent differences in the anesthetic dispersion patterns depending on the injection technique used. In the case of intra-articular infiltrations of the metatarsophalangeal (MTP) joint, the contrast dispersion was consistently focal, well defined, and reproducible, which aligns with the closed and encapsulated anatomical nature of the joint. The articular capsule acts as a physical barrier that contains the injected volume in a controlled manner, allowing the anesthetic effect to remain confined exclusively within the joint cavity. This containment facilitates precise diagnostic interpretation, as any clinical improvement can be attributed with a high degree of certainty to the blockade of articular structures, without interference from nervous or bursae tissues. In contrast, infiltrations performed in the intermetatarsal space—following the conventional pathway of the nerve suppression test—showed broad, irregular, and variable contrast dispersion, both distally and proximally. This behavior corresponds to the lack of anatomical encapsulation in this region, where the anesthetic can freely spread through soft tissues including fat, fascia, bursae, nerves, and articular branches. Such unpredictable diffusion carries a high risk of unintentionally anesthetizing adjacent structures, particularly the plantar articular branches, which can lead to nonspecific or falsely positive clinical responses.

The functional comparison between both techniques suggests that the MTP joint, due to its enclosed anatomical nature, provides an optimal environment for controlled diagnostic infiltration. In contrast, the intermetatarsal space, being anatomically loose and lacking defined boundaries, does not allow the anesthetic effect to be reliably isolated, thus complicating the subsequent clinical interpretation. These observations support the notion that the MTP joint provides an ideal setting for anatomically selective suppression testing, whereas neuroma infiltration—although clinically useful—may result in less specific responses due to variable anesthetic dispersion and possible coexistence of adjacent bursitis or fibrosis.

## 4. Discussion

Diagnosing metatarsal pain remains one of the greatest challenges in foot and ankle practice due to overlapping symptoms among multiple entities, including Morton’s neuroma, mechanical metatarsalgia, intermetatarsal bursitis, and plantar plate rupture. The findings of this experimental cadaveric study—based on anatomical assessment of contrast dispersion following targeted infiltrations—offer a functional rationale to improve differential diagnosis and potentially reduce therapeutic errors. However, it must be emphasized that these results are limited to contrast behavior and do not directly reflect the pharmacodynamics of local anesthetics, which should be addressed in future studies.

In this study, no communication was observed between the metatarsophalangeal (MTP) joint capsule and the intermetatarsal bursa, reinforcing the concept of the MTP joint as a sealed anatomical cavity. In contrast, the intermetatarsal space represents an open anatomical region, where injected fluid can spread extensively along connective tissue planes in a tunnel-like fashion. Anatomically, this diffusion may reach the plantar articular branches innervating the MTP capsule, even before these branches divide from the medial or lateral plantar nerves, which could explain false-positive results when interdigital infiltrations are performed. For this reason, quantifying average dispersion values in the intermetatarsal space was not feasible in our model, as the injected solution did not remain confined to a measurable compartment [12,13].

This new technique constitutes a reproducible and minimally invasive tool to optimize the differential diagnosis between mechanical metatarsalgia and Morton’s neuroma, thereby improving therapeutic selection and potentially reducing treatment failures (Table 1). Morton’s neuroma results from entrapment of the common plantar digital nerve, which courses between the third and fourth metatarsals and divides to innervate the adjacent toes. As described by Sarrafian and confirmed by classical anatomical studies (Sarrafian), this nerve does not provide direct articular branches to the MTP capsule. In contrast, the capsule receives innervation from plantar articular branches originating proximally from the medial and lateral plantar nerves, along with dorsal contributions from the superficial peroneal nerve [13]. This anatomical separation supports the rationale that MTP joint infiltrations may selectively block articular pain without affecting a distal neuroma. In contrast, as previously noted, interdigital infiltrations may still produce misleading relief due to fluid dispersion.

Comparative ultrasound-guided injections into the MTP joint and the intermetatarsal space provide a functional approach to identify the true source of pain. In our study, contained contrast dispersion in the MTP joint confirms its nature as a sealed cavity—ideal for precise diagnostic blockade. In contrast, the diffuse contrast pattern in the IM space reflects an open anatomy, making selective blockade of the interdigital nerve more difficult. This may explain the therapeutic failures seen with techniques such as radiofrequency ablation or neurectomy.

One of the most frequent errors is assuming that the presence of a neuroma on MRI or ultrasound always explains the patient’s pain. This structural misattribution—without functional confirmation—may result in unnecessary surgeries or ineffective treatments. Ruiz Santiago and Lucas et al. have shown that diagnostic infiltrations can alter surgical indications in up to 30% of cases [3,16]. Similarly, procedures targeting the plantar plate or metatarsal overload will fail if the pain originates from an untreated interdigital neuropathy. This underscores the need for a stepwise diagnostic strategy that integrates structural imaging, clinical examination, and functional tests such as those proposed here.

As evidence continues to grow around complex metatarsal pain syndromes, it becomes increasingly imperative to adopt a structured diagnostic approach that combines targeted history-taking, expert physical examination, high-resolution imaging, and—crucially—functional testing with anatomical support, as described in this study. This integrated methodology not only enhances diagnostic precision but also reduces unnecessary interventions and improves functional outcomes and patient satisfaction.

This study presents certain limitations. First, it was conducted on cadaveric specimens using radiological contrast, whose physicochemical properties differ from those of local anesthetic agents. Contrast is less diffusive and more viscous, whereas anesthetics have greater capacity to disperse through tissues and, in anatomically more open compartments, may expand beyond the injection site, potentially altering diagnostic specificity [18,19]. Second, the assumption that the metatarsophalangeal joint acts as a sealed cavity does not always hold true in the clinical setting. Conditions such as plantar plate rupture, capsular degeneration due to chronic overload, or rheumatic disease may compromise joint tightness, reducing the reproducibility of the test [20,21]. Therefore, the present work should be regarded as an anatomical validation rather than a demonstration of clinical efficacy. Future in vivo studies will be required to determine whether these findings translate into reliable diagnostic performance and improved therapeutic outcomes in the management of metatarsalgia and Morton’s neuroma.

## 5. Conclusions

The metatarsophalangeal (MTP) joint generally behaves as a closed cavity, with predictable contrast dispersion that allows for a more specific diagnostic blockade. In contrast, the intermetatarsal (IM) space has an open anatomy in which contrast spreads more diffusely and erratically, hindering the accurate diagnosis of Morton’s neuroma. However, in certain clinical conditions—such as plantar plate rupture, capsular degeneration, or rheumatic disease—the tightness of the joint cavity may be compromised. In this context, ultrasound guidance provides the opportunity to perform diagnostic injections that may significantly reduce the rate of failed diagnoses. A positive response to intra-articular injection—i.e., pain relief following MTP blockade—suggests an articular origin or involvement of the plantar plate. Conversely, an ambiguous or limited response suggests that the pain may be neuropathic in origin, in which case a diagnostic injection targeting the neuroma would constitute the next step for confirmation.

## Figures and Tables

**Figure 1 reports-08-00211-f001:**
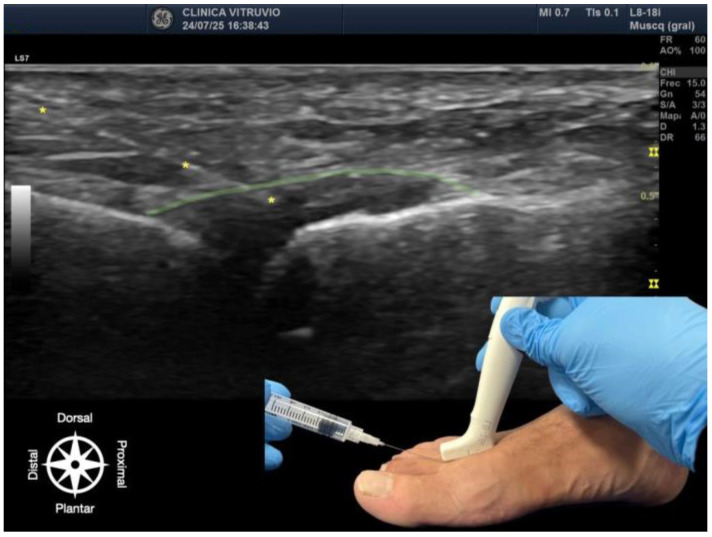
Ultrasound-guided intra-articular infiltration of the metatarsophalangeal joint (Dorsal view). The joint capsule (dorsal recess) is highlighted in green, showing the needle, marked with an asterisk (*) in yellow, inside the articular space. In the lower right corner, the dorsal approach with probe and needle placement is displayed. The anatomical orientation diagram indicates the proximal–distal and dorsal–plantar axes.

**Figure 2 reports-08-00211-f002:**
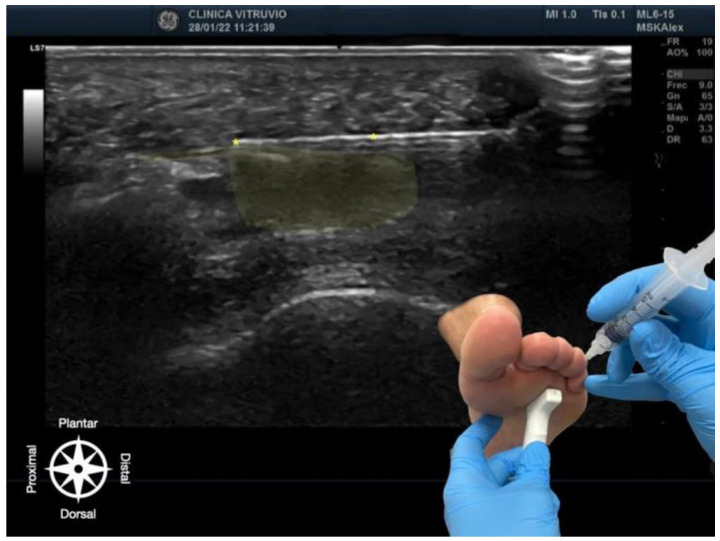
Ultrasound-guided infiltration of the third intermetatarsal space for Morton’s neuroma. The bursa–nerve complex is highlighted in yellow, with the needle following its path, also marked with yellow asterisks (*). In the lower right corner, the plantar approach is shown, illustrating probe and needle positioning. The anatomical orientation diagram indicates the proximal–distal and dorsal–plantar axes.

**Figure 3 reports-08-00211-f003:**
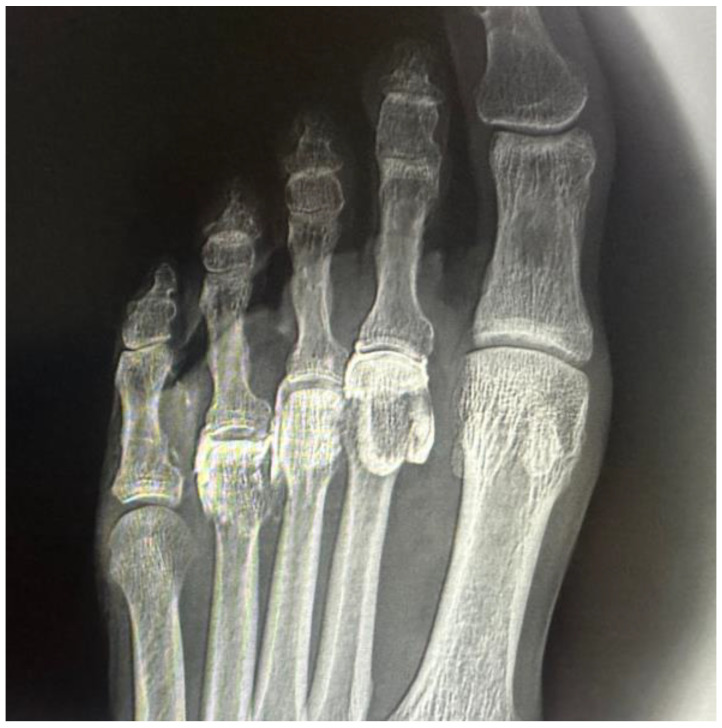
Dorsoplantar radiograph of the right forefoot following intra-articular injection of radiopaque contrast. A volume of 0.5 cc was injected into the second metatarsophalangeal joint, 0.4 cc into the third, and 0.3 cc into the fourth. Complete filling of the joint cavities is observed without contrast extravasation, confirming the sealed capsular nature of each articulation.

**Figure 4 reports-08-00211-f004:**
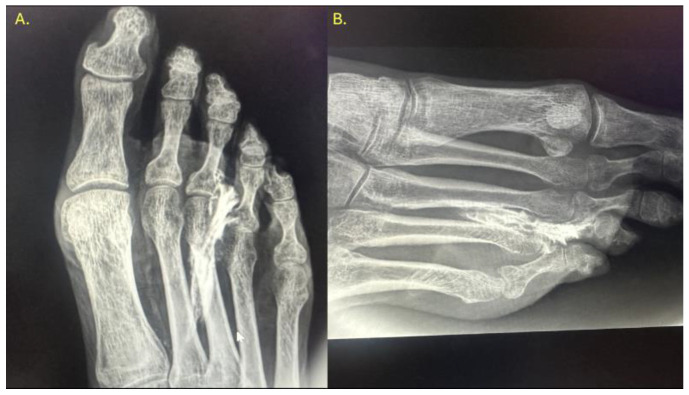
Ultrasound-guided injection of 0.5 cc of radiopaque contrast targeting the third common plantar digital nerve. The injection was performed in long axis, with a distal-to-proximal trajectory parallel to the nerve. (**A**) Dorsoplantar projection of the right forefoot. (**B**) Oblique projection of the same injection, allowing for better visualization of the contrast tracking along the interdigital space.

**Table 1 reports-08-00211-t001:** Key differential diagnostic features between mechanical metatarsalgia and Morton’s neuroma.

Feature	Mechanical Metatarsalgia	Morton’s Neuroma
Pain location	Plantar region under metatarsal heads	Burning pain in intermetatarsal space, radiating to adjacent toes
Pain trigger	Weight-bearing, overload, inappropriate footwear, limitation of ankle dorsiflexion (equinus)	Tight shoes, forefoot compression, prolonged walking; may be aggravated by biomechanical overload such as ankle equinus
Associated findings	Plantar plate degeneration/rupture, hallux valgus, long/short metatarsals	Perineural fibrosis, interdigital nerve thickening
Neurological symptoms	Not usually frequent; generally related to mechanical overload	Paresthesias, “electric shock” sensations, numbness
Imaging findings	Plantar plate tears, metatarsal overload, bursitis	Nerve thickening, intermetatarsal mass on US/MRI
Response to infiltration	Relief after intra-articular MTP injection	Lack of relief after intra-articular MTP injection, but improvement after interdigital nerve block
	Relief after intra-articular MTP injection	Lack of relief after intra-articular MTP injection, but improvement after interdigital nerve block

Legend: Comparative features between mechanical metatarsalgia and Morton’s neuroma. Adapted from anatomical and clinical descriptions [5,12,16].

## Data Availability

The original contributions presented in this study are included in this article. Further inquiries can be directed to the corresponding author.

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
