# Peer review of "Anatomical Validation of a Selective Anesthetic Block Test to Differentiate Morton’s Neuroma from Mechanical Metatarsalgia"

_reports, 2025, doi:10.3390/reports8040211_

Round 1

Reviewer 1 Report

Comments and Suggestions for Authors

Thank you to the authors for this work.
Thank you to the editor for this opportunity to review.
Great work, simple, clear, effective... with understandable clinical implications... in a very specific field, that is.
The context is excellently presented.
The written anatomical explanations are clear.
The differential diagnoses seem comprehensive.
The method seems correct.
Only a few comments: could you supplement your article with various diagrams/photos clearly illustrating the puncture techniques and an anatomical diagram for greater clarity during reading?
The same goes for the differential diagnoses, which would benefit from a small comparative table?
Congratulations!
Please keep me informed of the progress of this article. Thank you!

Author Response

Dear Reviewer 1. Thank you for appreciate our work. The changes in the paper due to your suggestions are highlighted in yellow

We have modified one photo and added a new one, illustrating the puncture techniques and an anatomical diagram for greater clarity during reading.

Also, ee have added a small comparative table about the differential diagnoses between both diseases.

Reviewer 2 Report

Comments and Suggestions for Authors

The article presented for review is aimed at an important diagnostic and clinical problem - differential diagnostics of metatarsalgia, more specifically - determining the probable role of Morton's neuroma in the forefoot pain. As the authors noted in the introduction to the article, visualization of nerve thickening on ultrasound or, especially, on MRI does not give an unambiguous answer about the role of this formation as a direct source of pain in a particular patient, which can lead to the wrong treatment. The authors noted that the nerve block in the intermetatarsal space cannot be considered a sufficiently effective method of differential diagnostics, since the anesthetic can spread towards the joint, causing its block, which can lead to a false positive interpretation of the test result. As an alternative, the authors proposed using a local block "from the opposite", that is, blocking the joint itself, assuming that its capsule is hermetic, and the contrast will not spread beyond it, including in the direction of the neuroma. The authors conducted an elegant cadaveric study, in which contrast was injected into the joint cavity and intermetatarsal space instead of contrast, and then its spread was assessed. As a result, it was shown that the injection of contrast into the intermetatarsal space is accompanied by its significant spread, including towards the joint, and the injection of contrast into the cavity of the second, third, and fourth metatarsophalangeal joints did not lead to a similar effect, while the contrast remained within the joint cavity. On this basis, the authors suggested the possibility of using a local anesthetic for an intra-articular block as a diagnostic for pain caused by Morton's neuroma "from the opposite". In general, the idea of ​​the study seems quite reasonable, it was carried out quite precisely, and, undoubtedly, is of interest to specialists involved in the diagnosis and treatment of the forefoot disorders. Nevertheless, it is important to note a number of limitations of the study, mainly related to the interpretation of its results. 
The title of the article sounds very attractive - "Anesthetic Suppression Test to Differentiate Morton's Neuroma from Mechanical Metatarsalgia: A new Diagnostic Tool?". After reading it, one can assume that the authors offer their own diagnostic test, study patients with metatarsalgia and Montan's neuroma, and also use an anesthetic. But reading the text of the article is somewhat disappointing - the study was performed on cadavers, contrast was used instead of an anesthetic, the study material was selected based on the principle of an intact forefoot, and had nothing to do with either neuroma or metatarsalgia (which is often accompanied by a desintegrity of the joint structures). The authors do not present the main thing - a diagnostic test with an assessment of its value (sensitivity, specificity, etc.). The results of the study are presented very briefly - the authors noted that in no case of injection into the joint was extravasation obtained, and in all cases of injection into the intermetatarsal space, significant distribution of the contrast was obtained. Of course, the reviewer has no reason to assume that these data (especially those concerning the accuracy of injection into the joints and the absence of leakage) are not true, but such high accuracy is somewhat alarming. As for the injection of contrast into the intermetatarsal space, it would be better to see at least average figures for the spread of contrast in different directions from the injection point in the scientific article. 
As the authors point out in the introduction, there can be many sources of pain in the forefoot and metatarsalgia. One cannot be sure that the injection of anesthetic into the joint will exhaust all possible options, except for Morton's neuroma (inflammation or damage to the capsule, surrounding tissues, etc. are possible). In addition, both rheumatic and chronic overload conditions can be accompanied by a lack of capsule tightness, which does not provide confidence in the reproducibility of this principle (and not the test, since the test was not demonstrated!). Thus, in fact, the authors conducted a study of the spread of contrast (not anesthetic!) in an experimental cadaver study. This is the recommended focus if the authors modify the article. All possible interpretations of these findings as potential for further clinical studies should be included in the discussion section with appropriate limitations.

Author Response

Dear Reviewer 2. Thank you for appreciate our work. The changes in the paper due to your suggestions are highlighted in green

We have change the title of the paper, reflecting better the aim of the study. 

We worked hard in the discussion section, limitations of the study and conclusions to adapt the paper following your suggestions. 

Reviewer 3 Report

Comments and Suggestions for Authors

Dear authors, 

Thank you for your submission on ‘Anesthetic Suppression Test to Differentiate Morton’s Neuroma from Mechanical Metatarsalgia: A new Diagnostic Tool?’

This is an interesting cadaveric study with potential clinical application. It is rooted in Anatomy which I do not think you have championed enough in the abstract or introduction as the basis / rationale for the study. 

As a result, your introduction is heavy on diagnostic difficulties and differentials with long lists when really it needs to be re-written to reflect the anatomy you are about to test and why you believe testing it may have a clinical application differentiating intraarticular versus neurological pain. 

I believe it has the potential of clinical use and therefore with major revisions should be reconsidered for publication. Comments below. 

Abstract

-Remove “So” in Background and objectives. 

-This needs revising to include the anatomical basis of the study to better / more easily inform the reader. 

Introduction

-Metatarsalgia and Morton’s neuroma clearly explained with adequate references. 

- Differential diagnoses made clear with presentation of argument for diagnostic difficulties with adequate references. 

- You seem to be diverging here on the challenge of diagnosis being due to either the anatomical difficulty of nerve injection blockade versus the sheer number of possible diagnoses. 

-  “The differential diagnosis is particularly challenging due to the convergence of symptoms” needs to start on a new paragraph. 

- “ Sarrafian and Kelikian provide a foundational description of the neural and articular arrangement of the forefoot (13). The MTP joints are innervated by dorsal and plantar articular branches arising from the superficial and deep peroneal nerves, as well as the medial and lateral plantar nerves. These articular branches emerge proximal to the formation of the common plantar digital nerve, implying that an intra-articular anesthetic injection should not block a neuroma located distally in the intermetatarsal space” This anatomical study forms the basis of your study. I would find a way to work this into the abstract as including this makes a lot more sense to the basic themes of the study in reading the abstract. 

- Well reasoned paragraph on imaging modalities. Again this needs to start on a new paragraph. 

- Page 3 “ This type of strategy could evolved beyond diagnostic application” The english language needs to be revised. 

Methods:

2.2 US guided technique explained clearly. 

2.3 Use of independent observation is good and adds to quality of study. 

2.5 Ethical considerations sound. 

Results:

 Figure 2- good XR of intrarticular joint infiltration efficacy. 

3.1 Good description of infiltration appearances post administration.

Figure 3- excellent XR of intermetatarsal infiltration. 

3.3 Well written and really defines the results and what you have been able to show. I would find a way to include this in the abstract along the lines of intra-articular infiltration is extremely well demarcated on XR.

Discussion 

Overall fairly reasoned however does not flow particularly well and would suggest restructuring and re-writing this section entirely. 

  • You need to note the limitations- this is a cadaveric study on a very limited number

  • What do you know about the spreading/infiltrative capabilities of short acting local anaesthetic crossing the joint capsule? Unlike XR contrast, you need to find out and reference that your produced methodology will remain sound when using a different medium, ie. local anaesthetic.You describe a sealed cavity but you need to discuss the action of anaesthetic agents and if there is any risk of diffusion through capsule/membranes.  I think you develop this to some extent in the third paragraph but this needs expanding. 

  • What are the next steps from this study? How would you envisage further testing in vivo and clinically? What would your group like to see happen for the next steps to push what you have done along further?

Conclusion

“The MTP joint behaves as a closed cavity with predictable contrast dispersion, allowing for effective diagnostic blockade. This contrasts with the open anatomy of the intermetatarsal (IM) space, where contrast spreads more diffusely and erratically, hindering accurate diagnosis of Morton’s neuroma” This is a good summary and should be included in abstract to better explain the background behind your study. 

Line 4 “A correct new ultrasound”  Remove ‘Correct’

Comments on the Quality of English Language

Generally fair. Some grammatical errors and unnecessary words have been included which require removal as discussed and noted above. 

Author Response

Dear Reviewer 3. Thank you for appreciate our work. The changes in the paper due to your suggestions are highlighted in blue

We worked hard in the introduction section to adapt the paper following your suggestions, giving a more detailes anatomically point of view 

Round 2

Reviewer 2 Report

Comments and Suggestions for Authors

The authors revised the article in accordance with the previous recommendation. The article is recommended for publication in the present version.

Author Response

Thank you for your suggestions, that made the paper better

Reviewer 3 Report

Comments and Suggestions for Authors

Dear authors, 

Thank you for returning your revised manuscript onAnatomical Validation of a Selective Anesthetic Block Test to Differentiating Morton’s Neuroma from Mechanical Metatarsalgia.’

You have followed reviewers comments and feedback well. 

Overall there are some corrections in English and grammar but the requested changes are a significant improvement on the first submission and this should be accepted for publication following minor grammatical changes. 

Title: Anatomical Validation of a Selective Anesthetic Block Test to Differentiating Morton’s Neuroma from Mechanical Metatar-salgia should be changed to ‘differentiate’

Abstract: 

Changes made to the introduction and the last paragraph of results highlight much more clearly the goals and messages of the paper. Well done. 

Introduction:

Paragraph two and three have been re-written and re-structured. This is a much more flowing introduction that highlights the scope of potential diagnoses and challenges. 

As an alternative, we propose a technique ́from the opposite ́ grammatically incorrect- needs changing to ‘ ‘Alternatively, we propose an opposing technique’ 

The following additional paragraph flows well with a concise and clear explanation and the final paragraph reads well in narrating the potential diagnostic and therapeutic applications.

Materials and Methods:

Figures 1A and 2 are well put together and highlight the technique and a snap shot of the US which is interesting and important to the reader. 

Results:

No new changes, comments or suggestions to make for this section. Well balanced and presented with good figures. 

Discussion:

Thank you for taking the time and effort to restructure this section of your paper. The overall flow and message is much clearer. 

-Potential pharmacodynamic challenges with different mediums has been noted and referenced. 

-Table 1 is clearly presented 

- The conclusion is clear and reads well. 

Comments on the Quality of English Language
  1. Title: Anatomical Validation of a Selective Anesthetic Block Test to Differentiating Morton’s Neuroma from Mechanical Metatar-salgia should be changed to ‘differentiate’
  2. Introduction section: As an alternative, we propose a technique ́from the opposite ́ grammatically incorrect- needs changing to ‘ ‘Alternatively, we propose an opposing technique’ 

Author Response

Dear Reviewer 3, thank you for your suggestions. The changes made to your review are underlined in the text in green. 

Thank you for returning your revised manuscript onAnatomical Validation of a Selective Anesthetic Block Test to Differentiating Morton’s Neuroma from Mechanical Metatarsalgia.’

You have followed reviewers comments and feedback well. Overall there are some corrections in English and grammar but the requested changes are a significant improvement on the first submission and this should be accepted for publication following minor grammatical changes. 

- Thank you for your tougths about the paper

Title: Anatomical Validation of a Selective Anesthetic Block Test to Differentiating Morton’s Neuroma from Mechanical Metatar-salgia should be changed to ‘differentiate’

- The change has been made in the text. the correct one now is "Anatomical Validation of a Selective Anesthetic Block Test to Differentiate Morton’s Neuroma from Mechanical Metatarsalgia"   

Abstract: 

Changes made to the introduction and the last paragraph of results highlight much more clearly the goals and messages of the paper. Well done. 

- Thanks !!

Introduction:

Paragraph two and three have been re-written and re-structured. This is a much more flowing introduction that highlights the scope of potential diagnoses and challenges. 

- Thanks !!

As an alternative, we propose a technique ́from the opposite ́ grammatically incorrect- needs changing to ‘ ‘Alternatively, we propose an opposing technique’ 

- This change has been made in the text. 

The following additional paragraph flows well with a concise and clear explanation and the final paragraph reads well in narrating the potential diagnostic and therapeutic applications.

- Thanks !!

Materials and Methods:

Figures 1A and 2 are well put together and highlight the technique and a snap shot of the US which is interesting and important to the reader. 

- Thanks !!

Results:

No new changes, comments or suggestions to make for this section. Well balanced and presented with good figures. 

- Thanks !!

Discussion:

Thank you for taking the time and effort to restructure this section of your paper. The overall flow and message is much clearer. 

- Thanks !!

-Potential pharmacodynamic challenges with different mediums has been noted and referenced. 

- Thanks !!

-Table 1 is clearly presented 

- Thanks !!

- The conclusion is clear and reads well. 

- Thanks !!

Comments on the Quality of English Language
  1. Title: Anatomical Validation of a Selective Anesthetic Block Test to Differentiating Morton’s Neuroma from Mechanical Metatar-salgia should be changed to ‘differentiate’
  2. Introduction section: As an alternative, we propose a technique ́from the opposite ́ grammatically incorrect- needs changing to ‘ ‘Alternatively, we propose an opposing technique’ 

- This two changes has been made in the text.